# Comparing the Efficacy of Multidisciplinary Assessment and Treatment, or Acceptance and Commitment Therapy, with Treatment as Usual on Health Outcomes in Women on Long-Term Sick Leave—A Randomised Controlled Trial

**DOI:** 10.3390/ijerph18041754

**Published:** 2021-02-11

**Authors:** Anna Finnes, Ingrid Anderzén, Ronnie Pingel, JoAnne Dahl, Linnea Molin, Per Lytsy

**Affiliations:** 1Division of Psychology, Department of Clinical Neuroscience, Karolinska Institutet, Nobels väg 9, SE-171 65 Solna, Sweden; 2Department of Public Health and Caring Sciences, Uppsala University, Husargatan 3, SE-751 22 Uppsala, Sweden; ingrid.anderzen@pubcare.uu.se (I.A.); per.lytsy@pubcare.uu.se (P.L.); 3Department of Statistics, Uppsala University, Kyrkogårdsgatan 10, SE-751 20 Uppsala, Sweden; ronnie.pingel@statistik.uu.se; 4Department of Psychology, Uppsala University, Campus Blåsenhus, von Kraemers allé 1A, SE-751 42 Uppsala, Sweden; jdahl8@me.com; 5Uppsala University Hospital, SE-751 85 Uppsala, Sweden; linnea.molin@akademiska.se; 6Division of Insurance Medicine, Department of Clinical Neuroscience, Karolinska Institutet, Berzelius väg 3, SE-171 65 Solna, Sweden

**Keywords:** rehabilitation, sick leave, acceptance and commitment therapy, multi-disciplinary rehabilitation, chronic pain, mental disorders, women

## Abstract

Background: Chronic pain and mental disorders are common reasons for long term sick leave. The study objective was to evaluate the efficacy of a multidisciplinary assessment and treatment program including acceptance and commitment therapy (TEAM) and stand-alone acceptance and commitment therapy (ACT), compared with treatment as usual (Control) on health outcomes in women on long-term sick leave. Method: Participants (*n* = 308), women of working age on long term sick leave due to musculoskeletal pain and/or common mental disorders, were randomized to TEAM (*n* = 102), ACT (*n* = 102) or Control (*n* = 104). Participants in the multidisciplinary assessment treatment program received ACT, but also medical assessment, occupational therapy and social counselling. The second intervention included ACT only. Health outcomes were assessed over 12 months using adjusted linear mixed models. The results showed significant interaction effects for both ACT and TEAM compared with Control in anxiety (ACT [*p* < 0.05]; TEAM [*p* < 0.001]), depression (ACT [*p* < 0.001]; TEAM [*p* < 0.001]) and general well-being (ACT [*p* < 0.05]; TEAM [*p* < 0.001]). For self-rated pain, there was a significant interaction effect in favour of ACT (*p* < 0.05), and for satisfaction with life in favour of TEAM (*p* < 0.001). Conclusion: Both ACT alone and multidisciplinary assessment and treatment including ACT were superior to treatment as usual in clinical outcomes.

## 1. Introduction

The leading causes of long-term sick leave are common mental disorders [1] and musculoskeletal pain [2], and co-morbidity between these disorders is common [3]. Multidisciplinary treatment following a biopsychosocial model is widely accepted as the “golden standard” in the treatment and rehabilitation offered to patients with chronic pain [4,5] and is provided by practitioners from two or more different disciplines collaborating in assessments, support and treatment. Evidence suggests that such integrated care is effective both in promoting return to work and in reducing disability and sick leave days due to both chronic pain and poor mental health [6]. However, there are mixed results regarding the effects of multidisciplinary treatment [7,8], which may be due to lack of consensus on how the work should be organised; programmes are diverse, with the timing, nature and intensity of the intervention varying considerably [8].

To further understand the added value of more than one modality in rehabilitation programs, single therapist interventions should be evaluated alongside multidisciplinary treatment. Additionally, it has been argued that in the context of patients on long-term sick leave for psychiatric and concomitant somatic disease, there is a need for individually tailored and flexible medical rehabilitation programmes [2].

Acceptance and commitment therapy (ACT) is a transdiagnostic psychological approach that allows for the conceptualization of common maintaining processes across present problems within one treatment protocol [9]. ACT has shown small to medium effect sizes on physical and mental health in chronic pain patients [10], as well as in the treatment of anxiety [11,12,13] and depression [12,13,14]. ACT has also been evaluated in transdiagnostic settings with patients on sick leave, albeit with inconsistent results [15,16].

The purpose of the present clinical trial was to investigate the effects of an ACT treatment programme and a multidisciplinary assessment and treatment programme tailored to meet individual needs (TEAM), also partly based on ACT, compared with treatment as usual (Control) in women on long-term sick leave due to musculoskeletal pain and/or common mental disorders. Both programmes involved coordinated rehabilitation efforts with those of the social security system. Long term sick leave because of musculoskeletal pain and/or common mental disorders is more common in women, and this study was designed to address this group in accordance with the aim of the overall funding programme. In a previous publication [17], we reported effects from this trial on return to work. Now, we wish to determine whether there were any differences in effects on health outcomes for the intervention groups compared with Control.

## 2. Materials and Methods

### 2.1. Design and Study Population

A three-armed randomized controlled trial was designed according to the CONSORT guidelines. The study population consisted of women on long-term sick leave due to pain and/or mental health disorders, recruited between and at the Swedish Social Insurance Agency in Uppsala County in Sweden. The study took place when new regulations were implemented in the social security policy, which implied an introduction of a maximum time restriction of reimbursed days with sickness benefits (refunding a maximum of 365 days in a 450-day period). When the maximum number of days were reached, unemployed persons on sick leave were transferred from the health insurance system to the employment insurance system to have their functioning and work ability reassessed.

Participants were assessed in two steps prior to inclusion. First, women on sick leave due to pain and/or mental illness that would be affected by the new regulations were identified in registers at the Swedish Social Insurance Agency. Figure 1 illustrates the flow chart of the inclusion, exclusion and data collection process. In the second step, the medical sick leave certificates were assessed regarding the inclusion and exclusion criteria.

### 2.2. Inclusion and Exclusion Ciriteria

Inclusion criteria were the following: being of working age (20–64 years), on long-term sick leave due to musculoskeletal pain and/or common mental disorders, approaching the maximum limit of sickness benefits. Exclusion criteria included the following: active suicide ideation; active alcohol/substance abuse or dependence; history of major mental disorders, i.e., psychosis, bipolar disorder type I; severe social dysfunction/personality disorder; currently involved in psychotherapy or other structured vocational rehabilitation programme, according to information found in the doctors’ certificates. The inclusion procedure has been described in more detail elsewhere [17].

### 2.3. Randomization Procedure

The participants were included consecutively and were randomised, into even triplets, by an administrator (who was not part of the research group) at the central Social Insurance Agency to one of three groups: (1) ACT-treatment with a psychologist, (2) multidisciplinary assessment and treatment (TEAM) or (3) treatment as usual (Control). However, some participants had to be excluded after randomisation due to either being enrolled prior to formal ethical approval, severe somatic or psychiatric disease in accordance with the exclusion criteria or being involved in a concurring rehabilitation programme or withdrawing their consent.

### 2.4. Interventions

#### 2.4.1. Collaboration with Other Authorities

In line with the new policy regulations, persons who reached the maximum days of sick leave compensation were transferred from the Social Insurance Agency to the Swedish Public Employment Services to have their work ability assessed [18]. The “work life introduction programme” consisted of an intensified assessment period aiming at identification of individual needs for support to return to work. The program targeted individuals that had been absent from the labour market for 2.5 years or more and lasted for a period of a maximum of three months. The work ability assessment included, e.g., meetings with an employment service officer, work training placements, meetings with employers, etc. After the introductory assessment, participants either continued activities within the employment service or, if the assessment showed that the individual had no work ability, applied for a new period of sickness benefits.

The participants in the two intervention groups received either treatment with acceptance and commitment therapy (ACT) or multidisciplinary assessment and treatment (TEAM). Both interventions were tailored to meet individual needs and were to last a maximum of twelve months. In both intervention arms, a professional involved in the participants’ treatment plan participated in scheduled collaboration meetings with the public employment service officer, representing the project. The purpose was to contribute to the work ability assessment from a medical rehabilitation perspective, inform on how symptoms can affect work disability and to set adequate expectations and goals for return to work.

#### 2.4.2. Acceptance and Commitment Therapy

Assessment and treatment in the ACT treatment arm were conducted by a psychologist. Assessment consisted of a clinical interview, including psychiatric diagnostic screening with the Mini-International Neuropsychiatric Interview (M.I.N.I.; Swedish Version 6.0.0d) [19]. An ACT protocol was created for this trial, with the aim to increase the participants’ psychological flexibility around the obstacles of returning to work specifically and of a better life quality in general. For all participants, treatments started with a behaviour analysis, where avoidant behaviour patterns underlying the current situation were investigated, together with a mapping of valued directions, i.e., valued areas of life, such as relationships, leisure time, work and self-caring activities such as physical activity that might have been put on hold due to attempts to control symptoms. Other modules of the protocol included acceptance/exposure exercises, which entailed learning new ways of accepting discomfort; perspective taking, or learning to take perspective on thoughts, feelings and sensations; self-compassion, which entailed learning to practice self-acceptance and self-care behaviours; and valued direction, which aimed at learning to increase and engage in behaviour in a variety of meaningful directions. The ACT sessions were mostly held at the clinic, but all participants were offered at least one session outside the clinic at a location of their choice, i.e., at home or at work. The purpose of this part of the protocol was to supplement the behaviour analysis with situations that were as real to the participant’s everyday life as possible. If a participant was not able to come to the clinic due to pain or other disability, further sessions outside the clinic were offered. Sessions were typically about 1 h long. The therapists providing ACT were all licensed clinical psychologists under the regular supervision of a peer reviewed ACT trainer.

#### 2.4.3. Multidisciplinary Assessment and Treatment

The multidisciplinary team consisted of a physician, a psychologist, an occupational therapist and a social worker. Weekly team meetings provided a forum for the clinicians to discuss new assessments as well as follow up on patients in treatment. The protocol for this treatment arm consisted of two phases: assessment and treatment. The assessment phase constituted the core of the intervention and started with each team specialist conducting an in-depth interview (1.5–2 h) with the participant. When the four interviews were completed, the case was discussed at a multidisciplinary team meeting focusing on how symptoms influenced the participant’s function, what factors contributed to continued disability, what resources and challenges could be identified and the team’s overall assessment of work ability and probability of return to work. The purpose of the team conference was also to establish an individualised rehabilitation plan that included actions from one or more team specialists with the common goal of improving possibilities to successfully participate in the work life introduction programme at the Public Employment Service. The plan included suggested interventions from one or more team specialists and sometimes suggestions of referral for further assessment and treatment elsewhere. The content in the assessment and treatment phases are further described in Appendix A. The participant had the choice of accepting the whole, none or parts of the suggested plan. Treatment started when the participant had agreed on the treatment plan. During the treatment phase, each case was discussed as needed at weekly team meetings, which provided an opportunity to evaluate the progress, to synchronise the planned or ongoing activities for each participant or to adjust the plan. The overall aim was that all treatment offered by the team members should comply with basic ACT principles. Therefore, all team members participated in several ACT workshops. The therapists providing ACT, all licensed clinical psychologists, also received regular supervision from a peer-reviewed ACT trainer.

#### 2.4.4. Control Group

Participants in the Control group received treatment as usual, following the standard work life introduction programme at the Public Employment Service. They were not offered any assessment or treatment from the study. Participants were informed that they were free to seek any medical counselling or other treatment as they wished.

### 2.5. Measures

Baseline data were collected from sick leave certificates or records at the social insurance agency. The diagnosis stated on the sick leave certificates was used to classify participants’ main problem as either psychiatric, pain-related or both. The participants filled in questionnaires on three occasions: pre-measurement (0), after 6 months (0.5) and after one year (1).

Satisfaction with the treatment was evaluated by means of seven questions (if they had felt treated with respect, if their needs had been acknowledged, if they had felt involved in the treatment, overall satisfaction with the therapist/team specialists, if it fulfilled their needs, perceived quality of the treatment received, an overall estimation of satisfaction with having taken part in the study). Responses were given on a scale from 1 (not at all) to 4 (completely). In this sample, the Cronbach’s alpha for this scale was 0.924.

Level of pain intensity was measured by one separate question “Do you have pain? (yes/no)” from the Örebro Musculoskeletal Pain Questionnaire [20]. If the response was yes, the respondent was further asked to rate the mean pain level over the past three months on a scale ranging from 0 = no pain at all to 10 = unbearable amount of pain.

Depression and anxiety levels were measured with the Hospital Anxiety and Depression Scale (HADS) [21] to assess the symptom severity and caseness of anxiety disorders and depression [22]. HADS consists of 14 items and two subscales with a scale range of 0 to 21, respectively. A cut-off score of 8 has been established for both subscales [22]. In the current sample, the Cronbach’s alpha at baseline for the anxiety subscale was 0.87 and for the depression subscale 0.85.

The satisfaction with life scale (SWLS) was developed to measure general subjective satisfaction with life [23]. Five items are scored on a scale from 1 to 7. Low scores indicate low satisfaction with life, whereas high scores indicate high satisfaction with life. The SWLS is a validated instrument in research on well-being [24]. In this sample, Cronbach’s alpha was estimated at 0.90 at baseline

General well-being was assessed with the 12-item version of the General Health Questionnaire (GHQ-12) [25]. Total score range is 0 to 36 points, in which lower score indicates distress and a higher score indicates higher general well-being. The GHQ-12 has been widely validated and has high internal reliability [26,27]. Cronbach’s alpha was estimated at 0.85 for this scale in the current sample.

### 2.6. Statistical and Data Analysis

Differences in baseline characteristics between each intervention group and the control group were investigated using χ^2^ tests for proportions and *t*-tests for continuous data. The data were analysed according to the intention-to-treat (ITT) principle in order to provide information about the treatment policy effects. Multiple imputation by chained equations was used to handle missing data, due to loss of follow-up data. All measurement occasions and all baseline variables, except education and country of origin (excluded due to large amounts of missing data (13%)), were included in the imputation model, with predictive mean matching being used for quantitative variables and logistic/polytomous/proportional odds regression for categorical variables. The imputation was carried out 100 times. To study the impact of the imputation, the means of the outcome measures at each occasion were calculated both on the original data and the imputed data. In order to evaluate the effect of the treatments, three regression models were estimated. First, as a benchmark, a crude linear model was applied to the original data to estimate the differences in means at the last measurement occasion. The crude linear model did not include the other measurement occasions in the analysis and did not take into account missing data since it was carried out on the original data. Second, in order to improve efficiency and to take into account the missing data, we estimated a crude linear mixed model including all measurement occasions in the model. We specified the model so that the coefficients could be interpreted as the differences in means between the treatments and the control at the last measurement occasion. The crude linear mixed model included a random intercept to handle within individual correlation, and the analysis was carried out on the imputed data. Third, we added baseline characteristics (age, employment, diagnosis, years on sick leave, reimbursement) to the crude linear mixed model, yielding an adjusted linear mixed model. Differences between groups regarding treatment completion and satisfaction with the treatment were analysed with ANOVA. All tests were two sided, and *p* < 0.05 was considered statistically significant. The statistical analyses were performed using IBM SPSS Statistics Version 23 and R version 3.2.0 and the packages “lme4” [28] and “mice” [29].

The clinically significant change was evaluated from pre-measurement to 6 months and from pre-measurement to 12 months using the Reliable Change Index (RCI) with twofold criteria, as proposed by Jacobson and Truax [30]. Complete case analyses were performed, excluding any participants with missing data on the subscales, respectively. Participants who reported clinical levels of anxiety or depression at pre-measurement were included in the analysis. RCI was calculated using Swedish norms for the subscales of anxiety and depression from the HADS questionnaire. Cut-off scores, indicating a reliable change (the C criteria), were calculated for the HADS depression subscale (6.1) and for the HADS anxiety subscale (7.1). The C criteria were employed to examine whether participants had passed the cut off between the general and clinical population. Participants are classified as either recovered, improved, unchanged or deteriorated.

### 2.7. Ethical Considerations and Trial Registration

The study was registered retrospectively on 15 November 2017, at the Clinicaltrials.gov Register Platform (ID NCT03343457). The start date for recruiting participants was June 2010, ending in June 2011. The study was approved by the Regional Ethics Committee in Uppsala (Dnr 2010/088) following the Declaration of Helsinki for the ethical principles of medical research involving humans and was performed at a clinical vocational rehabilitation centre at the University hospital in Uppsala, Sweden, in collaboration with Uppsala University.

## 3. Results

### 3.1. Sociodemographic and Pre-Treatment Data

A total of 308 participants were included in the study. Figure 1 provides a flowchart of participants throughout the trial. There were no significant differences between the groups (Control, ACT and TEAM) on the sociodemographic variables or on the pre-treatment outcome measures (see Table 1), indicating homogeneity between groups. A more detailed description of the study groups’ composition and medical disorders have been published previously [17].

### 3.2. Treatment Duration and Satisfaction with Treatment

Participants rated how satisfied they were with the treatment they had received in the treatment groups. There were no significant differences in satisfaction with treatment between ACT (*M* = 3.3, *SD* = 0.62) and TEAM (*M* = 3.2, *SD* = 0.71), *F*(1, 96) = 0.067, *p* = 0.796. Regarding how helpful the collaboration effort with SIA and the SPES office had been in terms of facilitating RTW, there was no significant difference in ratings between ACT (*M* = 1.8, *SD* = 0.99) and TEAM (*M* = 1.7, *SD* = 0.89), *F*(1, 96) = 0.148, *p* = 0.701. Treatment time between ACT and TEAM was significantly different (*F*(1, 158) = 25.14, *p* = 0.000), where TEAM was the longer intervention. Further data on treatment specification can be found in Appendix A.

### 3.3. Clinical Efficacy

Means and standard deviations of study variables at pre-treatment, 6 months and 12 months are presented in Table 2. For means on imputed study variables, please refer to Appendix A. There were significant differences in means in favour of both ACT and TEAM compared with Control at 12 months (see Table 3). Further, mixed effect models for repeated measures analysis (time×group interactions) are shown in Table 4. The adjusted mixed effects model for repeated measures analysis with imputed data identified significant changes over time in the severity of pain ratings in favour of ACT, in symptoms of anxiety and depression as well as in satisfaction with life in favour of both ACT and TEAM, and in ratings of general well-being in favour of TEAM as compared with Control after 12 months. Confidence intervals for these models are displayed in Figure 2.

### 3.4. Missing Data

When missing data were taken into account, all outcome measures were adjusted upwards for pain, anxiety and depression in all treatment groups, suggesting that those who dropped out had more severe problems. Likewise, for satisfaction with life and general well-being, the means were higher in the original data compared to imputed data, suggesting that those with higher satisfaction with life and general well-being responded to questionnaires to a higher degree.

### 3.5. Clinically Significant Change

The clinically significant change for depression and anxiety is presented in Table 5. Overall, a majority of participants remained unchanged in all groups during the intervention year. However, there were consistently more participants with no change in the Control group. At 12 months, more patients had recovered in the TEAM intervention group for both anxiety and depression, compared to the ACT and Control groups. Few participants deteriorated, and these were more or less disseminated over all groups.

## 4. Discussion

This randomized controlled trial that investigated the effects of two different treatment programmes compared with treatment as usual found that both a multidisciplinary assessment and a treatment programme and ACT delivered by a single therapist led to greater improvement on health outcomes in persons on long-term sick leave due to musculoskeletal disorders and/or common mental disorders. Compared with the Control group, both ACT and TEAM reduced levels of anxiety and depression and increased satisfaction with life. ACT alone reduced levels of pain, and TEAM increased general well-being. This is of interest since the ACT intervention demands substantially less resources and may be more cost-effective. However, there are also obvious advantages of being able to offer a variety of interventions within a multidisciplinary setting, especially considering the complexity that is often the case with long sick leave spells. We expected a superiority of the TEAM intervention, considering the higher intensity and more resources both in assessing symptoms and work ability and more treatment obtained, but in terms of symptom improvement, this assumption was not fulfilled.

In a previous publication, the main outcome of the study, the effect on return to work was explored [17]. Both ACT and TEAM were superior to Control; however, overall the effect was stronger for TEAM participants. The common factor between all intervention groups was the work life introduction programme at the Swedish Public Employment Service. Both experimental groups included collaboration with the employment service and coordination of activities, which may have had an effect also on self-rated pain and mental health. The purpose of the collaboration between the employment service and the health care was to contribute to an assessment of functional ability from a medical and psychological perspective in the assessment of the work ability process. A professional involved in the participants’ treatment plan also participated in scheduled meetings at the employment service together with the participant. The aim was to provide guidance from a health care perspective in setting adequate goals for rehabilitation activities. Sometimes, activities within the project such as participating in therapy sessions were included in the vocational rehabilitation as a means to gradually increase work-related activities. Considering the long periods of sick leave that proceeded the intervention, this integrated approach including the support of a therapist may have contributed to less stress in the work ability assessment, which may be reflected in the outcome measures.

It may be assumed that several years on sick leave substantially affects health status and that returning to work may also be an important health indicator, especially in the context of these participants where work ability was evaluated for the first time in many years. Although the collaboration effort was present in both treatment options, the TEAM intervention offered more opportunities for work-focused interventions given the involvement of an occupational therapist who did workplace visits, evaluated ergonomics, etc. Earlier results on this study population have shown a significant increase in working hours per week for the TEAM intervention [17]. This was not the case for the ACT intervention, which provided less opportunities for a work focus, consisting of more traditional clinical psychotherapy. Later trials where, for example, cognitive behaviour therapy is integrated with a work focus and more comprehensive return to work programmes have been effective in both reducing symptoms and enhancing return to work for patients on long-term sick leave [31].

The unimodal and the multidisciplinary interventions were received equally well by the patients, as indicated by scores on satisfaction with treatment. More patients in the ACT intervention completed the treatment phase according to the plan; however, the treatment phase for participants in the TEAM group was preceded by a rather intense assessment procedure, and further intense treatment plans may have been perceived as too much. Adding up those who completed at least one of the treatment modalities suggested, there were no differences between groups with regard to completion. However, for mean treatment duration, the TEAM intervention was longer than the ACT intervention and also more intense in terms of number of sessions compared with ACT. One circumstance in relation to this is that the multidisciplinary team assessed the need for interventions as somewhat larger than what was actually delivered in the end. This may be either an expression of less accuracy in the assessment or by saturation from the participants’ side. Some participants expressed that they could not manage a more intense treatment plan and preferred to focus on one treatment modality.

When clinically significant change scores were evaluated, even though most participants did not reach a clinically significant change regardless of group, more participants in TEAM were categorised as recovered 12 months after inclusion, compared to both ACT and the Control group. This may be an expression of the TEAM intervention being more flexible and more tailored to personal needs, owing to a more thorough assessment procedure. Considering the extremely long periods of sick leave for the participants in the present study, it is reasonable to assume that maintaining factors for depression vary extensively between participants. Again, the wider range of available interventions in TEAM may have helped more participants; however, for some participants, a single-therapist approach to treatment may be sufficient and more cost-effective.

There were some important limitations with this trial. First, due to a procedural mistake, the participants were aware of the results of the allocation procedure before they filled in the pre-measurements. Although there were no significant differences between groups for the outcome measures at pre-measurement, participants in the Control group consistently rated more pain and symptoms of mental ill-health compared with ACT and TEAM. It may have been perceived as missing out on valuable help and support in difficult circumstances that may have contributed to further distress, which may be reflected in the ratings. Second, there was a high number of participants that never showed up or dropped out, as well as missing data. This might have produced a selection bias, where persons with good effects remained in the programme, whereas those not having an effect were left out. We addressed this problem by using multiple imputations; however, it was observed that the means in the imputed data are adjusted upwards or downwards compared with the crude means depending on the theory, i.e., participants that are worse off fail, to a greater degree, to respond to the questionnaire at 0, 6 and 12 months.

## 5. Conclusions

The main conclusion of this study is that both single-therapist ACT treatment and multidisciplinary assessment and treatment based on ACT were separately superior to treatment as usual in reducing symptoms of pain and mental disorders and increasing satisfaction with life and general well-being. This study adds evidence that important aspects of health and wellbeing may be improved in women with a very long history of sick leave. There is a need for further research to confirm these findings in other settings and in other populations, such as men, and also to clarify differences in efficacy between single-therapist and multidisciplinary treatment. In addition, research should address the “core components” needed to produce the effect, for example, in designs where the effect of further treatment is examined when added to psychological treatment. There is also a need to assess and evaluate costs and cost effectiveness, alongside clinical trials, as interventions demand different resources and may differ in duration and intensity.

## Figures and Tables

**Figure 1 ijerph-18-01754-f001:**
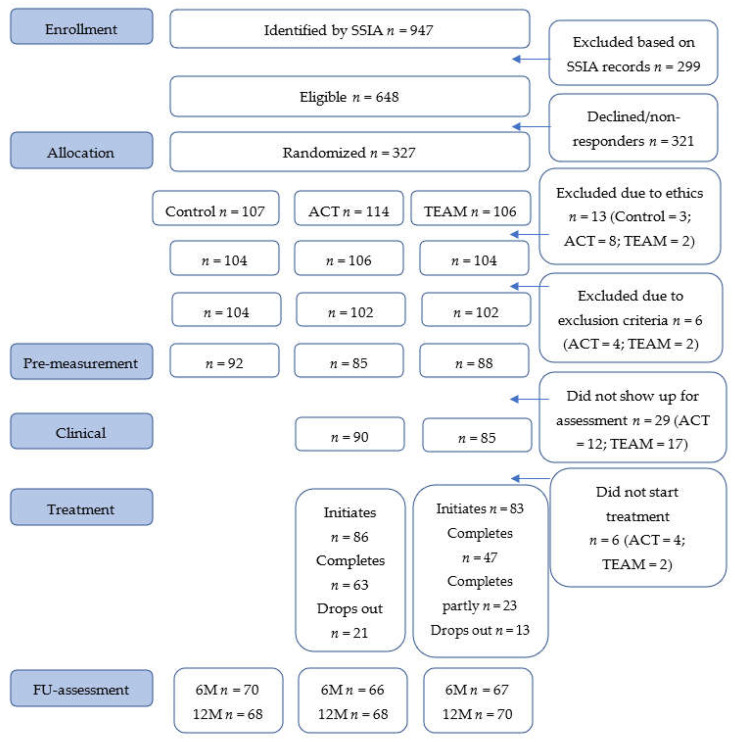
Flowchart of the study.

**Figure 2 ijerph-18-01754-f002:**
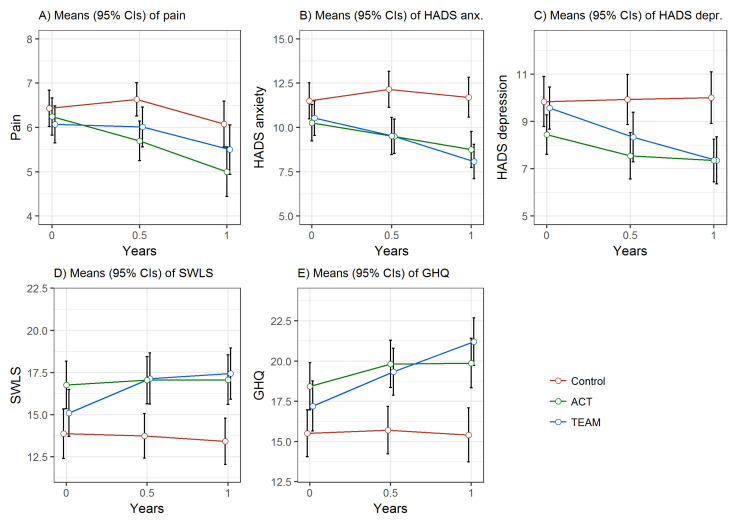
Mean outcomes on imputed data of (**A**) pain, (**B**) anxiety, (**C**) depression, (**D**) satisfaction with life and (**E**) general well-being at 0, 6, and 12 months for ACT, TEAM and Control with 95% confidence intervals. Note: ACT—acceptance and commitment therapy; TEAM—multidisciplinary assessment and treatment programme; HADS—hospital anxiety and depression scale; SWLS—satisfaction with life scale; GHQ—general health questionnaire.

**Table 1 ijerph-18-01754-t001:** Baseline characteristics of the participants.

	Control	ACT	TEAM
	*n* = 104	*n* = 102	*n* = 102
Age mean (sd)	47.5 (8.34)	47.8 (7.76)	49.9 (8.74)
Education *n* (%)			
Primary school	21 (22.8)	17 (19.5)	24 (27.3)
Secondary school	39 (42.4)	37 (42.5)	38 (43.2)
University education	32 (34.8)	33 (37.9)	26 (29.5)
Born outside Sweden *n* (%)	22 (24.2)	17 (19.5)	15 (16.9)
Unemployed *n* (%)	37 (35.6)	43 (42.2)	33 (32.4)
Diagnosis *n* (%)			
Psychiatric diagnosis	31 (29.8)	41 (40.2)	27 (26.5)
Pain diagnosis	39 (37.5)	34 (33.3)	34 (33.3)
Psychiatric and pain diagnosis	34 (32.7)	25 (24.5)	40 (39.2)
Other diagnosis	0 (0.0)	2 (2.0)	1 (1.0)
Years of sick leave (mean (sd))	7.52 (3.36)	7.57 (3.15)	7.50 (3.08)
Reimbursement *n* (%)			
25%	13 (12.6)	15 (14.7)	12 (11.9)
50%	31 (30.1)	23 (22.5)	33 (32.7)
75%	5 (4.9)	9 (8.8)	8 (7.9)
100%	54 (52.4)	55 (53.9)	48 (47.5)

**Table 2 ijerph-18-01754-t002:** Number of respondents, means, and standard deviations for the original data at baseline, 6 and 12 months for the outcome measures.

	Baseline		6 Months		12 Months	
Pain	*n*	*M (SD)*	*n*	*M (SD)*	*n*	*M (SD)*
ACT	71	6.08 (1.98)	43	5.67 (2.15)	33	4.63 (2.90)
TEAM	83	6.13 (2.05)	38	5.59 (2.11)	32	5.07 (2.77)
Control	88	6.53 (2.11)	38	6.83(1.67)	35	5.91 (2.65)
HADS anxiety						
ACT	19	10.05 (4.91)	35	9.00 (5.16)	34	8.70 (5.15)
TEAM	17	10.31 (4.79)	35	8.83 (4.78)	31	7.37 (4.51)
Control	12	11.09 (5.08)	33	11.37 (5.14)	35	10.71 (5.48)
HADS depression						
ACT	85	8.45 (4.19)	65	7.37 (5.04)	67	7.06 (4.67)
TEAM	86	9.22 (4.46)	66	7.70 (4.87)	70	6.56 (4.81)
Control	91	9.22 (5.13)	70	9.54 (5.22)	67	9.48 (5.11)
SWLS						
ACT	83	16.72 (7.05)	64	17.12 (7.53)	67	17.40 (7.91)
TEAM	88	15.26 (7.05)	67	17.70 (7.81)	69	18.94 (7.70)
Control	91	14.51 (7.48)	70	14.06 (6.61)	66	14.33 (6.92)
GHQ						
ACT	84	18.80 (7.50)	63	20.86 (7.52)	66	20.85 (7.80)
TEAM	87	17.25 (7.86)	67	20.40 (7.54)	66	22.20 (7.24)
Control	92	16.43 (7.14)	70	16.10 (7.22)	69	16.88 (8.27)

Note: ACT—acceptance and commitment therapy; TEAM—multidisciplinary assessment and treatment programme; HADS—hospital anxiety and depression scale; SWLS—satisfaction with life scale; GHQ—generalized health questionnaire.

**Table 3 ijerph-18-01754-t003:** Estimated differences in means for intervention groups compared with control at 12 months.

	Crude Linear Model on Original Data	Crude Linear Mixed Model on Imputed Data	Adjusted Linear Mixed Model on Imputed Data
	MD (CI)	MD (CI)	MD (CI)
Pain			
ACT	−1.28 ** (−2.22, −0.34)	−1.2 ** (−1.93, −0.48)	−1.16 ** (−1.85, −0.47)
TEAM	−0.84 (−1.77, 0.1)	−0.76 * (−1.5, −0.02)	−0.73 * (−1.45, −0.02)
HADS anxiety			
ACT	−2 * (−3.72, −0.29)	−2.54 ** (−4.17, −0.91)	−2.78 *** (−4.34, −1.23)
TEAM	−3.33 ** (−5.03, −1.64)	−3.5 *** (−5.1, −1.9)	−3.43 *** (−4.97, −1.89)
HADS depression			
ACT	−2.42 ** (−4.08, −0.76)	−2.74 *** (−4.28, −1.2)	−2.95 *** (−4.4, −1.5)
TEAM	−2.92 ** (−4.56, −1.28)	−2.86 ** (−4.38, −1.33)	−2.75 ** (−4.18, −1.32)
SWLS			
ACT	3.07 ** (0.5, 5.64)	3.68 ** (1.41,5.95)	3.97 *** (1.82, 6.12)
TEAM	4.61 *** (2.05, 7.16)	4.64 *** (2.38, 6.9)	4.52 *** (2.36, 6.67)
GHQ			
ACT	3.97 ** (1.3, 6.64)	4.45 *** (1.97, 6.94)	4.89 *** (2.55, 7.23)
TEAM	5.32 *** (2.68, 7.96)	5.56 *** (3.15, 7.96)	5.38 *** (3.1, 7.65)

Note: MD—difference in means; CI—confidence interval; ACT—acceptance and commitment therapy; TEAM—multidisciplinary assessment and treatment programme; HADS—hospital anxiety and depression scale; SWLS—satisfaction with life scale; GHQ—generalized health questionnaire. * *p* < 0.05; ** *p* < 0.01; *** *p* < 0.001.

**Table 4 ijerph-18-01754-t004:** Estimates of fixed effects for interaction effects regarding change over time for the experimental groups (time*group) as compared with the reference group Control for the outcome measures. Results from the non-adjusted model are presented as well as the results from the adjusted models.

	Pain		HADS Anxiety		HADS Depression		SWLS		GHQ	
	Crude	Adjusted	Crude	Adjusted	Crude	Adjusted	Crude	Adjusted	Crude	Adjusted
Intercept	6.45 ***	5.51 ***	11.24 ***	13.0 ***	9.24 ***	11.49 ***	14.44 ***	12.47 ***	16.34 ***	13.86 ***
Time	−0.21	−0.23	−0.02	0.05	0.68	0.71	−0.67	−0.75	−0.31	−0.36
ACT	−0.56	−0.51	−1.08	−1.63 *	−0.73	−1.22	2.34 *	2.96 **	2.62 *	3.46 ***
TEAM	−0.50	−0.49	−0.81	−0.80	0.15	0.15	0.93	0.87	0.89	0.79
Time*ACT	−0.87 **	−0.84 *	−1.49 *	−1.59 *	−2.29 ***	−2.34 ***	1.45	1.62	2.15 *	2.29 *
Time*TEAM	−0.49	−0.43	−2.71 ***	−2.82 ***	−3.29 ***	−3.29 ***	3.96 ***	3.96 ***	4.94 ***	4.92 ***
Age		−0.00		−0.02		−0.03		0.03		0.03
Employed		0.63 *		0.44		0.26		−1.03		−0.17
Diagnosis pain		1.25 ***		−3.40 ***		−2.95 ***		3.20 ***		5.11 ***
Diagnosis pain and psych		1.08 ***		−0.88		−0.87		0.15		2.44 *
Years sick leave		−0.03		0.02		−0.01		0.03		−0.08
Sick leave 50%		0.52		0.51		1.58 *		−0.57		−1.37
Sick leave 75%		1.29 *		0.72		2.27 *		−1.79		−2.01
Sick leave 100%		1.61 ***		2.61 **		4.11 ***		−4.68 ***		−5.54 ***

Note: ACT—acceptance and commitment therapy; TEAM—multidisciplinary assessment and treatment programme; HADS—hospital anxiety and depression scale; SWLS—satisfaction with life scale; GHQ—generalized health questionnaire. ** p* < 0.05; ** *p* < 0.01; *** *p* < 0.001.

**Table 5 ijerph-18-01754-t005:** Clinically significant change for anxiety and depression at 6 months and 12 months presented in percent in each category.

	ACT		TEAM		Control	
Months	6	12	6	12	6	12
HADS anxiety	*n* = 57		*n* = 62		*n* = 72	
Recovered	13	12	9	34	4	10
Improved	10	7	4	4	4	2
Unchanged	77	79	84	62	85	80
Deteriorated	0	2	2	0	8	8
HADS depression	*n* = 57		*n* = 60		*n* = 64	
Recovered	13	14	24	36	4	2
Improved	5	12	2	11	4	0
Unchanged	77	71	67	47	85	90
Deteriorated	5	2	7	7	6	7

Note: ACT—acceptance and commitment therapy; TEAM—multidisciplinary assessment and treatment programme; HADS—hospital anxiety and depression scale.

## Data Availability

The data presented in this study are available on request from the corresponding author. The data are not publicly available due to restrictions in data management stated in the informed consent.

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
