# Peer review of "Comparing the Efficacy of Multidisciplinary Assessment and Treatment, or Acceptance and Commitment Therapy, with Treatment as Usual on Health Outcomes in Women on Long-Term Sick Leave—A Randomised Controlled Trial"

_ijerph, 2021, doi:10.3390/ijerph18041754_

Round 1
Reviewer 1 Report
How to effectively treat to reduce the occurrence of long-term sick leave is an important issue for health maintenance. Especially breaking through the past treatment methods, the comparison of TEAM and ACT as treatment is more academically important. However, based on academic requirements, I made the following review comments.
- The authors take "woman" as the research object. However, the article did not explain why “woman” was chosen as the research object. Please add explanations from the authors.
- In "section 2.5", most of the scales do not provide scale range and reliability data, please add.
- In "section 2.7", the research data comes from 10 years ago. The source of the information is too old. Can the authors explain why they used materials from so long ago? In addition, because the source of the data is 10 years ago, the authors should also explain the research progress in the past 10 years in the “discussion section”.
- The authors mentioned: "Both ACT and TEAM were superior to Control however overall the effect was stonger for TEAM participants." Is "stonger" the wrong way to write "stronger"?
- In line 23, “N=308” should be expressed as “n=308”.
Author Response
We want to thank the reviewer for providing valuable feed-back and helping make the manuscript clearer. Below we respond to each comment separately:
- In Sweden women are over represented in long term sick leave regarding musculoskeletal pain and/or common mental disorders, and this study was designed specifically to address this group in accordance to the overall aim of a research program financed by the Swedish Ministry of Health and Social Affairs. We have now clarified this in the manuscript, please refer to line 65-68, in red font.
- We have added information on scale range and internal consistency of all scales. Please see section 2.5, changes in red font.
- Yes the reviewer is correct. The data was collected 10 years ago, however, the analyses were planned from the beginning and follows the original plan. Unfortunately, the analysis of data and writing process were delayed due to organizational changes. While this is not optimal, we believe that is does not affect the validity of the results.
- Yes, that is correct. The typo is corrected to "stronger" (line 338).
- We have changed “N” to “n” in line with the suggestion.
Reviewer 2 Report
A competently executed and written piece of research.
The results are of general interest, although the particular Swedish setting cannot perhaps be replicated everywhere. A little unusual perhaps to have two types of treatment evaluated at the same time, but I do not really see particular problem with that.
Author Response
We want to thank the reviewer for the positive feedback.
Reviewer 3 Report
As relevant aspects of the study we can mention that the Acceptance and Commitment Therapy and the evaluation and treatment proved to be more positive than the usual treatments used.
Strengths: characterization of the theme, methodology, inclusion and exclusion criteria, important limitations with this essay.
Weaknesses: the conclusions
Improvement suggestions
In the conclusions: "Future studies can be done by adding an additional treatment to the psychological treatment by examining the additive effects". They must substantiate them, as well as their relevance to this topic.
In the conclusions, the main conclusions of the study, the added value of the study for a better understanding of the topic and future investigations should be systematically described.
Author Response
We thank the reviewer for pointing out this obscurity in the manuscript. We have revised the conclusion section (please refer to lines 410 to 421, in red font) to clarify and better describe the relevance to the topic in line with the reviewers' suggestion.